# Beyond Amyloid: A Machine Learning-Driven Approach Reveals Properties of Potent GSK-3β Inhibitors Targeting Neurofibrillary Tangles

**DOI:** 10.3390/ijms25052646

**Published:** 2024-02-24

**Authors:** Martin Nwadiugwu, Ikenna Onwuekwe, Echezona Ezeanolue, Hongwen Deng

**Affiliations:** 1Tulane Center for Biomedical Informatics and Genomics, Deming Department of Medicine, Tulane University School of Medicine, Tulane University, New Orleans, LA 70112, USA; 2Neurology Unit, Department of Medicine, University of Nigeria Teaching Hospital, Ituku-Ozalla 400001, Enugu, Nigeria; ikenna.onwuekwe@unn.edu.ng; 3Department of Medicine, College of Medicine, University of Nigeria, Enugu Campus, Nsukka 400001, Enugu, Nigeria; 4Center for Translation and Implementation Research (CTAIR), University of Nigeria, Nsukka 410001, Enugu, Nigeria; eezeanolue@gmail.com; 5Healthy Sunrise Foundation, Las Vegas, NV 89107, USA

**Keywords:** Alzheimer’s disease, GSK-3β, neurofibrillary tangles, tau phosphorylation, machine learning

## Abstract

Current treatments for Alzheimer’s disease (AD) focus on slowing memory and cognitive decline, but none offer curative outcomes. This study aims to explore and curate the common properties of active, drug-like molecules that modulate glycogen synthase kinase 3β (GSK-3β), a well-documented kinase with increased activity in tau hyperphosphorylation and neurofibrillary tangles—hallmarks of AD pathology. Leveraging quantitative structure–activity relationship (QSAR) data from the PubChem and ChEMBL databases, we employed seven machine learning models: logistic regression (LogR), k-nearest neighbors (KNN), random forest (RF), support vector machine (SVM), extreme gradient boosting (XGB), neural networks (NNs), and ensemble majority voting. Our goal was to correctly predict active and inactive compounds that inhibit GSK-3β activity and identify their key properties. Among the six individual models, the NN demonstrated the highest performance with a 79% AUC-ROC on unbalanced external validation data, while the SVM model was superior in accurately classifying the compounds. The SVM and RF models surpassed NN in terms of Kappa values, and the ensemble majority voting model demonstrated slightly better accuracy to the NN on the external validation data. Feature importance analysis revealed that hydrogen bonds, phenol groups, and specific electronic characteristics are important features of molecular descriptors that positively correlate with active GSK-3β inhibition. Conversely, structural features like imidazole rings, sulfides, and methoxy groups showed a negative correlation. Our study highlights the significance of structural, electronic, and physicochemical descriptors in screening active candidates against GSK-3β. These predictive features could prove useful in therapeutic strategies to understand the important properties of GSK-3β candidate inhibitors that may potentially benefit non-amyloid-based AD treatments targeting neurofibrillary tangles.

## 1. Introduction

Approximately 44 million people worldwide are living with Alzheimer’s disease (AD), a devastating condition that currently has no cure and limited treatment options [1,2,3]. Existing medications can only slow memory and cognitive decline [4], leaving an urgent need for breakthrough therapies, as we are still unable to reverse cognitive impairment. Our research presents a promising pathway forward by utilizing extensive databases (PubChem and ChEMBL) and sophisticated machine learning algorithms to identify critical features of drugs that could potentially reverse cognitive decline by targeting glycogen synthase kinase 3β (GSK-3β)—a brain enzyme. Increased activity of GSK-3β in AD may lead to abnormal neurogenesis, compromised synaptic plasticity, and adverse effects in the hippocampus [5]. A primary therapeutic strategy for AD involves inhibiting GSK-3β because its activity is associated with the number of neurofibrillary tangles (NFTs) in the brains of AD patients [5,6]. The importance of exploring non-amyloid-based etiologies for AD, particularly in individuals with dementia who are free from amyloid plaques, cannot be overstated, and targeting GSK-3β for NFT reduction could offer a promising lead [7].

GSK-3 is predominantly located in axons; it is expressed in the central nervous system (CNS) and has alpha and beta isoforms, which are, respectively, coded by chromosomes 19 and 3 [5]. The beta isoform (GSK-3β) is the primary kinase responsible for tau protein phosphorylation; it can phosphorylate tau at 42 sites, inhibiting transcription factors such as heat shock transcription factor 1 to prevent cell toxic insults and modulate intrinsic pathways of cellular apoptosis favored by amyloid beta (Aβ) peptide [5,8,9,10]. GSK-3 also plays a role in modulating regulators involved in NFT formation; it has been reported to phosphorylate MyRF, a transcription factor that contributes to NFTs and participates in controlling the transcription of its target genes [7,11,12].

GSK-3β is associated with memory loss and learning impairment due to its role in increasing tau phosphorylation [5,6], which leads to tau disassembly from microtubules. This process results in cognitive impairment, axonal alterations, enhancement of long-term depression, and a decrease in long-lasting signal transmission between neurons by modulating N-methyl-D-aspartate (NMDA) receptors [5,13,14]. NMDA receptors in the CNS play a vital role in both synaptic transmission and plasticity, which are crucial for learning and memory [15]. The activation of GSK-3β through the phosphorylation of tyrosine 216 and serine 9 is regulated by signaling pathways such as PI3K/Akt and Wnt/β-catenin, which may be dysregulated in AD, leading to increased GSK-3β activity [5,13,16]. The inhibition of GSK-3β has been shown to be protective against NFT formation and neurodegeneration. For instance, in a splenectomized rat model, lithium inhibition of GSK-3β significantly reduced phospho-tau levels and protected against neurodegeneration, spatial learning, and memory deficits [17]. 

In this study, we focus on the inhibition of GSK-3β, a key factor in tau protein phosphorylation linked to AD pathology. We employ machine learning algorithms on active chemical compounds with inhibitory effects on GSK-3β to identify selective important chemical properties. The aim is to investigate the potential of drug-like molecules in modulating GSK-3β activity as a therapeutic strategy for AD treatments, targeting tau hyperphosphorylation and NFT formation. Our analysis seeks to understand the relationship between the chemical structures of GSK-3β inhibitors and their activity within the context of AD, utilizing a systems pharmacology framework and quantitative structure–activity relationship (QSAR) modeling, a technique used to discover connections between chemical compounds’ biological activities and their structural properties [18,19]. Indeed, regulatory agencies are increasingly employing QSAR models for predictive purposes, and to develop specialized tools and databases [20] with interpretable models using machine learning [21,22]. More effective drugs targeting AD have been predicted using a multi-target QSAR approach that achieved over 90% accuracy in classifying active and inactive compounds [23]. This surpasses the limitations of current single-target drugs by capturing a broader spectrum of drug-target interactions [23].

In brief, our research advances the identification of key properties of tested compounds that inhibit GSK-3β activity by examining structural features, incorporating electronic and physicochemical descriptors, and highlighting the impact of specific structural elements. This analysis not only provides new insights into GSK-3β inhibition but also paves the way for future investigation aimed at finding characteristics of more potent and effective inhibitors.

## 2. Results

Using PubChem IDs, we selected and downloaded only those assays that contained an active compound testing for GSK-3β inhibition, totaling 38 assays. From each bioassay, we extracted a PubChem chemical ID (CID), the Canonical Simplified Molecular Input Line Entry System (SMILES), and the activity outcome (active or inactive). The selected assays comprised a total of 3093 tested compounds, including 308 active, 2475 inactive, 73 unspecified, and 237 inconclusive chemical compounds, as detailed in Appendix A. We eliminated duplicate chemicals and excluded those with unspecified and inconclusive outcomes, resulting in a refined dataset of 225 active and 1212 inactive compounds for further data filtering and manipulation. Given the typical imbalance in bioassays which favor more inactive than active outcomes [18], our dataset exhibited similar characteristics with fewer active compounds. To counteract this, we employed a deliberate strategy to balance the dataset by selecting an equal number of active and inactive compounds (225 each) for modeling. This approach ensured a balanced representation of outcomes for more accurate and unbiased model training and validation. 

We generated a graphical representation of the frequency of different standard values (IC50) within the assay dataset (found in Appendix A) to analyze their distribution and variability, and to understand the nature of the assay. IC50 values explains the quantitative relationship between a substance’s concentration and the corresponding biological response measured in the assay [24]. As a relative measure of inhibitory potency, IC50 values facilitate the comparison of different substances or treatments’ effectiveness within a specific experimental context [25]. The histogram revealed that the concentrations followed a normal distribution, indicating that the selected compounds converged around a common standard. The diagram showing the distribution of log transformed IC50 values is presented in Figure 1B. 

To ensure a robust outcome, we used a balanced dataset of 450 compounds (see Appendix A) containing all unique 225 active compounds and 225 inactive compounds randomly selected from the total 1212 inactive compounds. Our decision for the random selection was driven by the need to create a representative sample, ensuring a balanced representation of active and inactive compounds. This approach is intended to mitigate bias and improve the generalizability of our models across diverse chemical spaces. Next, we randomly divided the dataset into two parts: 70% of the dataset was used as a training set, and the other 30% was used as validation (15%) and testing sets (15%), respectively. Additional validation was performed using a combined new external dataset of 433 compounds from two PubChem bioassay records (651569, 493225). The external validation data (EVD) (refer to Appendix A) contained 23.8% active compounds (103 active and 330 inactive); the machine-generated molecular descriptors for each compound were also obtained. The target variable, ACTIVITY (1 = active, 0 = inactive), represents the compounds’ efficacy in inhibiting or modulating GSK-3β. We chose the target variable as the key factor in the correlation analysis (Figure 2C) because it directly relates to the central objective of our study, which is to identify relationships between molecular descriptors and the biological activity of compounds against GSK-3β. It represents the biological response or effect of each compound, making it a critical measure for our analysis. By correlating it with molecular descriptors, we explored strongly associated features and obtained insights into the characteristics of potentially effective compounds. 

Next, we compared the ensemble weighting method by majority voting with the six individual models: the combined machine-generated molecular descriptors (data with QSAR descriptors in RDKit) and six types of machine learning methods (NN, RF, LogR, SVM, XGB, KNN). As shown in Table 1, NN exhibited the best model prediction for active and inactive compounds across all datasets with a 79% AUC–ROC score on the unbalanced external validation data, which contained only 23.8% active compounds. This was followed by SVM and RF (both at 72%) and the ensemble method (71%). While the NN method had the best AUC–ROC score, the SVM and RF achieved higher Kappa values (Table 1), indicating NN is likely less robust given the significant difference in prediction outcome between the validation dataset and the training and test sets.

In terms of learning methodology, RF appeared more robust and less prone to overfitting, as evidenced by the consistency of its predictions. The AUC–ROC and accuracy scores were less likely to fluctuate in the test and validation sets. Similar scores were observed in the SVM and ensemble models. The SVM and RF outperformed the NN model in terms of accuracy and demonstrated superior Cohen’s Kappa values on the external validation data. Cohen’s Kappa is a statistical measure that quantifies the extent to which the observed agreement (accuracy) between predicted and actual labels exceeds what would be expected by random chance [26]. It accounts for both actual and predicted accuracies, offering a more comprehensive assessment of a model’s predictive capability than accuracy alone [26]. LogR, KNN, and XGB exhibited the lowest AUC–ROC scores (69%) on the new external validation data.

Furthermore, the results showed that SVM and RF models exhibited slightly superior predictive robustness compared to the majority voting ensemble model. While RF and the majority voting models are forms of ensemble methods, they differ significantly in their approach to combining predictions from individual models. Majority voting uses a simple and straightforward method to aggregate model predictions and works well for diverse models; but offers limited potential for improvement. In contrast, RF is a more complex ensemble method that learns and optimizes the combination of predictions, often leading to higher accuracy [27,28]. Research has shown that the predictive efficacy of ensemble learning surpasses that of any single learner when the individual algorithms are both accurate and diverse [29]. RF is considered an ensemble method that relies on multiple decision trees to mitigate the risk of overfitting seen in individual decision trees and is considered the benchmark for evaluating the performance of new prediction methods and the gold standard in QSAR prediction [18].

### 2.1. Strengths and Limitations of Individual and Majority Voting Ensemble Models

The performance of the seven models based on their confusion matrices showed that LogR, KNN, and XGB exhibited similar limitations with a slight bias towards false positives, while RF identified true negatives but struggled with false positives. The NN model had a good balance, with lower false negatives compared to other models, while SVM showed the best overall performance in correctly identifying both active and inactive compounds, as shown in Figure 2B.

The observed performance differential between the majority voting ensemble method and the single models can be attributed to the specific characteristics and sensitivities of the individual models to the dataset’s features. The ensemble method’s performance on the EVD, at 71%, suggests that while it can leverage the strengths of individual models, it may also inherit their limitations, particularly in handling false positives and negatives [30]. The slightly lower performance could stem from an averaging effect, where the model-specific advantages in recognizing critical molecular descriptors are diluted. For example, SVM’s emphasis on structural descriptors and RF’s focus on electronic and physicochemical descriptors highlight their unique approaches to parsing the dataset. These models leverage specific types of features to predict GSK-3β inhibition outcomes accurately. The ensemble method, by combining these diverse approaches, may not capitalize on the specific strengths of each model, particularly when those strengths are derived from leveraging deep model-specific feature sets for prediction. This explanation aligns with the understanding that ensemble methods can improve prediction accuracy when individual models are both accurate and diverse [31] but may perform sub-optimally if the combination dilutes the distinct advantages of its constituent models or fails to harmonize their insights effectively [30].

### 2.2. Important Features

We selected and visualized the top 20 significant features from among 212 molecular descriptors used in our analysis, specifically for SVM, logR, RF, and XGB models, as illustrated in Table 2. The significant features for logR and RF were determined using their coefficients and feature importance scores, respectively. Similarly, XGB’s gradient boosting framework of feature importance measures is accessible and comparable to RF. Due to the inherent limitations of the KNN model, we were unable to generate a plot for its significant features. The KNN depends on distance metrics and lacks direct methods for evaluating the importance of individual features in its model output [32]. The top features identified by logR are easily interpretable, as shown in Figure 2A. LogR is a generalized linear model that utilizes the logistic function to transform linear predictions into probabilities for binary classification, in contrast to the nonlinear kernel of the SVM (RBF), as well as the methodologies of RF and XGB models. The SVM, logR, and RF models each consistently identified the same top important features across their respective runs. This stands in sharp contrast to the NN model, where the results regarding feature importance varied significantly between runs. The inconsistency in feature importance identification by NN could be attributed to its diverse architectures [18], particularly the use of dropout in our study. NNs are often considered “black boxes,” making it challenging to understand how feature importance is determined, unlike more transparent linear models [33]. This “black box” nature and the architectural differences in NNs are likely the causes of the observed variability in the feature importance plots across different runs. To address this, methods such as SHAP (Shapley additive explanation) and LIME (local interpretable model-agnostic explanation) have been developed. These techniques are designed to clarify how each feature contributes to the overall prediction in black box models [34].

Furthermore, we compared the important features from SVM, logR, RF, and XGB as presented in Table 2. These features were organized into distinct groups based on their properties, roles, and characteristics. Features that provide insight into the molecular framework, functional groups, and overall architecture of a molecule were classified as structural descriptors. Those that describe the electronic properties of a molecule, including charge distribution, electronic states, and molecular orbitals, were identified as electronic descriptors. Key variables related to the physical and chemical properties of a molecule, such as size, shape, and solubility, were categorized as physicochemical descriptors. Furthermore, features that offer a comprehensive overview of a molecule’s overall properties, often integrating information from structural, electronic, and physicochemical perspectives, were grouped as molecular descriptors. As shown in Table 2, most of the top features identified by SVM were primarily structural descriptors, while logR attributed more importance to both structural and electronic descriptors. The significant variables for RF were mainly characterized by electronic and physicochemical descriptors, whereas XGB placed greater emphasis on molecular descriptors in its model permutation.

The analysis of SVM and RF features, as shown in Table 2, reveals that the top five significant features identified by SVM are related to phenol, oxygen-containing compounds, and aromatic rings. This observation may suggest that compounds featuring aromatic hydroxyl groups, phenol groups, and phenol groups not engaged in ortho-hydrogen bonding may be positively correlated with effective GSK-3β inhibition. Furthermore, the top five features identified by the RF model are associated with molecular topological polar surface area, electronegativity, the electronic properties of atoms within a molecule, and molecular fingerprints indicative of substructures within the molecule.

A close look at the shared key features in logR and SVM models (Figure 3) showed that compounds containing nitrogen and oxygen atoms positively correlate with GSK-3β inhibition. This observation suggests that the inclusion of these specific atoms may enhance the compounds’ efficacy against GSK-3β. In contrast, the presence of structural descriptors, such as an imidazole ring, sulfide, and methoxy groups within a compound, is negatively associated with GSK-3β inhibition, indicating that these compounds may not effectively inhibit GSK-3β. Notably, methoxy groups were highlighted as significant features in the SVM, logR, and XGB models (as listed in Table 2). Additionally, compounds with lower polarizability of atoms (Avglpc) are less likely to correlate positively with active outcomes against GSK-3β. Understanding these structure–activity relationships can aid in developing more effective GSK-3β inhibitors.

## 3. Discussion

The inhibition of GSK-3β is considered crucial in treating AD because it has been shown to lead to a decrease in phosphorylated tau [35,36]. Recent studies have found that GSK-3β inhibitors can alleviate cognitive impairments associated with AD, prompting the initiation of clinical trials for potential drug candidates [37]. In our study, we identified hydrogen bonds, phenol groups, nitrogen, and oxygen atoms as significant structural characteristics of compounds inhibiting GSK-3β activity with positive outcomes. Previous research have highlighted the importance of hydrogen bonding and the presence of phenol groups in designing ligands to stabilize the DFG-out conformation of GSK-3β through a type II inhibition mechanism [37]. DFG-in and DFG-out refer to alternative conformations of the activation loop in protein kinases, which are associated with the active and inactive states of the GSK-3β kinase, respectively [38]. In the DFG-in conformation of the human GSK-3β enzyme, hydrogen bonds formed between specific residues—Lys103, Gln206, and Phe175—prevented the formation of the DFG-out conformation, keeping the kinase active [37]. Our study found that phenol groups that are not involved in ortho-hydrogen bonding were important features correlated with active outcomes of GSK-3β inhibitors (Figure 3 and Table 2).

GSK-3β inhibitors have neuroprotective effects, but designing a potent compound is challenging due to the active conformation (DFG-in), where the conserved DFG motif is oriented toward the ATP binding site, a site for which many have found challenging to design an inhibitor with good selectivity and potency [37,39,40]. Our study suggests that the charge distribution, molecular orbitals, electronic states, and electrostatic interactions of a molecule could be considered in screening active candidates against GSK-3β. Electronic descriptors were important characteristics commonly generated in the SVM, RF, logR, and XGB models of our study. For instance, our results show that the PEOE_VSA family of descriptors ranks among the top 20 important features in the four individual machine learning models (SVM, RF, logR, and XGB). PEOE_VSA is a van der Waals surface area (VSA) descriptor designed to capture direct electrostatic interactions [41]. While the specific PEOE_VSA type varied across models, with PEOE_VSA6, PEOE_VSA11, and PEOE_VSA12 showing a positive correlation with active outcomes compared to PEOE_VSA3, the PEOE_VSA descriptor was crucial in distinguishing relatively small differences in a congeneric series of compounds. Additionally, classical electrostatic attraction (hydrogen bonds) has been reported as a key feature in type II GSK-3β inhibition [37]. This attraction exists between charged or polar groups and specific amino acid residues, such as hydrogen bond attractions with the DFG Loop Asp (Aspartic Acid), glutamic acid residue in the αC-helix, and the hinge region of the GSK-3β kinase enzyme [37]. The αC-helix is a structural element in GSK-3β enzyme, while the hinge region is the flexible segment connecting the N- and C-terminal lobes of the kinase [42].

Furthermore, previous work has suggested that the underlying atomic contributions to partial charge and molar refractivity are relevant to receptor affinity [43]. We found that properties of compounds with lower polarizability of atoms (Avglpc) are less likely to be associated with active outcomes against GSK-3β. Polarizability relates to an atom’s ability to undergo induced dipole interactions and can influence the structural characteristics associated with the DFG-out conformation. Vijayan et al. [38] have suggested the distance between two specific pairs of atoms in the DFG motif and the αC-helix for a classical DFG-out conformation. The influence of polarizability on the DFG-out conformation can impact the activity of compounds inhibiting GSK-3β, highlighting that specific molecular conformation and structural features play a significant role not only in shaping the kinase’s conformation but also in determining interactions with ligands.

In our study, we did not observe a positive correlation between active GSK-3β inhibition outcomes and the presence of certain structural descriptors, such as an imidazole rings, sulfide, and methoxy groups. However, the precise arrangement of methoxy phenyl rings, particularly at the fourth carbon relative to the substituent’s attachment point (para-substitution), has been recognized in other studies to influence the binding efficacy of micromolar hit compounds that inhibit human GSK-3β and their nuanced structural differences [37]. This emphasizes the importance of specific structural features in binding affinity.

Our study acknowledges several limitations. First, the absence of a positive correlation with specific structural descriptors does not necessarily negate their importance; rather, it highlights the inherent complexity of the relationship between molecular structure and GSK-3β inhibition. Additionally, we have not explored potential synergistic effects or interactions between multiple structural features, which could influence compound activity. While our findings offer valuable insights into the role of structural, electronic, and physicochemical descriptors in compounds that are active or inactive against human GSK-3β, further studies are necessary to establish a definitive consensus on descriptors and properties correlating with active GSK-3β inhibition. Understanding how different substituents or modifications impact the reactivity and bioactivities of GSK-3β inhibitors remains a critical area for future research. Building upon existing research, this study not only enhances our knowledge of GSK-3β inhibition but also opens exciting avenues for developing more potent and effective inhibitors in the future.

## 4. Materials and Methods

### 4.1. Obtaining the Dataset

We searched for the UniProt ID (P49841) corresponding to GSK-3β and retrieved the 2D chemical structure (Figure 1A) along with associated ChEMBL database information (CHEMBL262). A comprehensive query of the ChEMBL database yielded a total of 2803 assays related to all bioactivity associated with GSK-3β. Our primary focus was on obtaining IC50 values from binding assays, which measure the efficacy of molecules in inhibiting GSK-3β. Additionally, we gathered information regarding the bioactivity experiments and the PubChem open chemistry database IDs [44]. The chemical structures of the tested molecules were provided in the SMILES format.

### 4.2. Generating Descriptors of Chemical Compounds

We generated molecular descriptors for the selected compounds based on their SMILES representation using the RDKit library in Python, an open-source toolkit for cheminformatics [45]. The RDKit descriptors are reported to exhibit high predictive ability, with higher R^2^ and lower RMSE in regression analysis compared to other fingerprint-type descriptor sets [46]. Molecular descriptors quantitatively represent various molecular properties such as structure, size, shape, and other characteristics that can be used to characterize chemical compounds. Our generated descriptors encompass a range of standard RDKit descriptors, including molecular weight, the number of valence electrons, and calculated properties related to distribution and permeability. These calculated properties include the logarithm of the partition coefficient (logP), atomic logP for lipophilicity (ALOGP), and the logarithm of the brain-to-blood partition coefficient (LogBB).

### 4.3. Machine Learning Analyses

Our dataset underwent rigorous cleaning and preprocessing to ensure the data’s quality and consistency for model training and validation. We assessed the datasets for missing values and employed mean imputation for molecular descriptors with missing values. The target variable (ACTIVITY) was scrutinized for missing entries, which were removed to maintain data integrity. We applied feature standardization to normalize the scale of molecular descriptors and to promote more stable and faster convergence in our machine learning models.

We utilized the Keras library (integrated with TensorFlow version 2.7.0) for neural network implementation, the Scikit-learn library (version 1.3.0) for machine learning methods, and Pandas (version 1.5.3) and NumPy (version 1.24.2) for data manipulation. For plotting, we used Matplotlib (version 3.4.1) and Seaborn (version 0.12.2), and we used RDKit (version 2023.03.3) to prepare the input representation of chemical compounds. Our approach included a plain feed-forward neural network (NN) and five conventional machine learning methods: random forest (RF), logistic regression (LogR), support vector machine (SVM), extreme gradient boosting (XGB), and k-nearest neighbor (KNN). Additionally, we employed majority voting to combine the predictions of the activity class (0 or 1) to which a chemical compound belongs from these individual machine learning methods.

The NN model comprised of three fully connected dense feed-forward layers: The first layer, with 128 units and a rectified linear unit (ReLU) activation function, is followed by a second layer with 64 units and a ReLU activation function. To prevent overfitting, a dropout layer with a rate of 0.5 was strategically placed after the first and second dense layers. The final layer, designed for binary classification, features a single unit with a sigmoid activation function. We employed the Adam optimizer [47] and utilized a binary cross-entropy loss with a learning rate of 0.001, 10 epochs, and a mini-batch size of 32.

Our analysis used QSAR molecular descriptors, leading to the development of seven models that incorporated all unique combinations of machine learning algorithms (NN, RF, LogR, SVM, XGB, KNN) and molecular descriptors (QSAR descriptors from RDKit). The SVM model, which employed a radial basis function (RBF) kernel, was configured with a penalty parameter of 1.0, which is the default setting. Similarly, both the XGB and RF models were set up with 100 estimators, reflecting their default configurations optimized through grid search and experimental efficiency. The prediction probabilities generated by these machine learning methods were then utilized in majority voting, a form of ensemble learning [18] designed to potentially boost the overall prediction performance. Ensemble learning encourages a synergy among a diverse set of models, enabling them to reach a consensus decision through majority voting [48]. In this process, individual predictions from each model are transformed into binary outcomes (0 or 1) based on a predefined threshold (≥3). For every data point, these binary predictions are aggregated, and if the total reaches or exceeds the threshold, the ensemble prediction is designated as 1; otherwise, it is set to 0. The final prediction is determined by majority decision of the individual models. 

The evaluation metrics include accuracy, recall, and area under the curve–receiver operating characteristics (AUC–ROC). Hyperparameter tuning and sensitivity analysis were used to assess and balance trade-offs between complexity and interpretability. The rationale is to find a model that strikes a balance in terms of complexity, performance, and interpretability. A 10-fold cross-validation process was used to validate the models, dividing the data into 10 segments and training the model 10 times, each time excluding a different segment to evaluate its performance [49]. The Python script used in the analysis is available online as Jupyter notebook in the GitHub repository.

## 5. Conclusions

In summary, while effective GSK-3β inhibitors are important for AD treatment, substantial work is required to develop effective compounds with refined designs and characteristics. Our study underscores the importance of structural, electronic, and physicochemical descriptors in screening active candidates against GSK-3β. Our machine learning NN model achieved 79% accuracy in classifying active and inactive compounds on external validation data with an unbalanced dataset and revealed important features such as hydrogen bonds, phenol groups, and specific electronic characteristics playing crucial roles in the model permutation. Future studies should explore how these specific features affect compound reactivity and efficacy in inhibiting GSK-3β, to expand our understanding of molecular attributes that govern GSK-3β inhibition. Such research will not only improve our current predictive models but also potentially reveal new molecular entities that could act as versatile inhibitors for GSK-3β, and perhaps other targets related to AD and similar neurodegenerative conditions. Additionally, exploring synergies between fragment-based descriptors and our existing methodologies, could propel the discovery of novel GSK-3β inhibitors, to contribute to the overarching goal of developing more effective therapeutic strategies for AD.

## Figures and Tables

**Figure 1 ijms-25-02646-f001:**
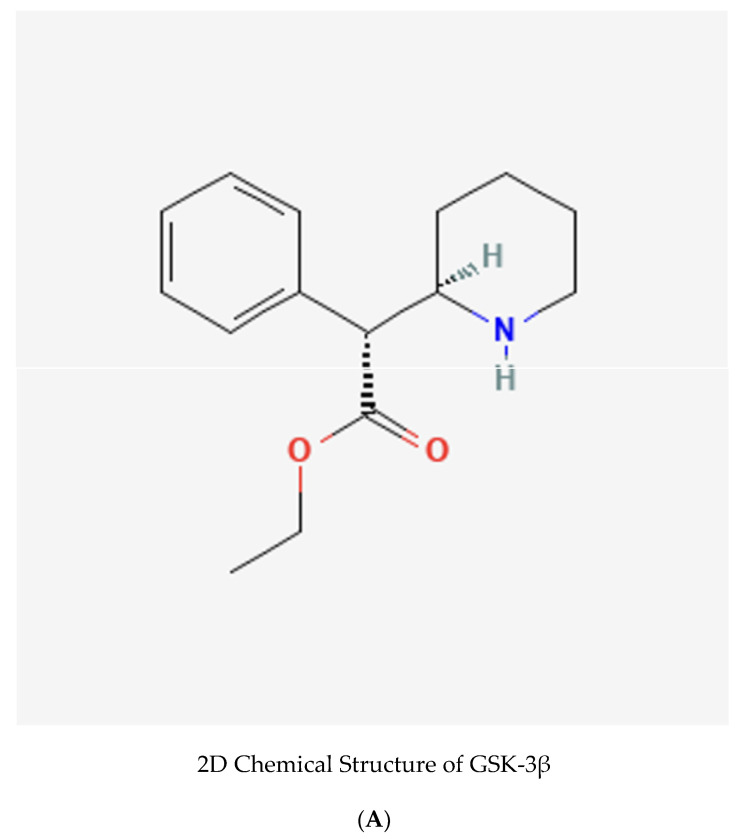
(**A**) 2D chemical structure of GSK-3β. (**B**) Histogram showing the distribution of IC50 values. Selected compounds mainly converged around a similar standard.

**Figure 2 ijms-25-02646-f002:**
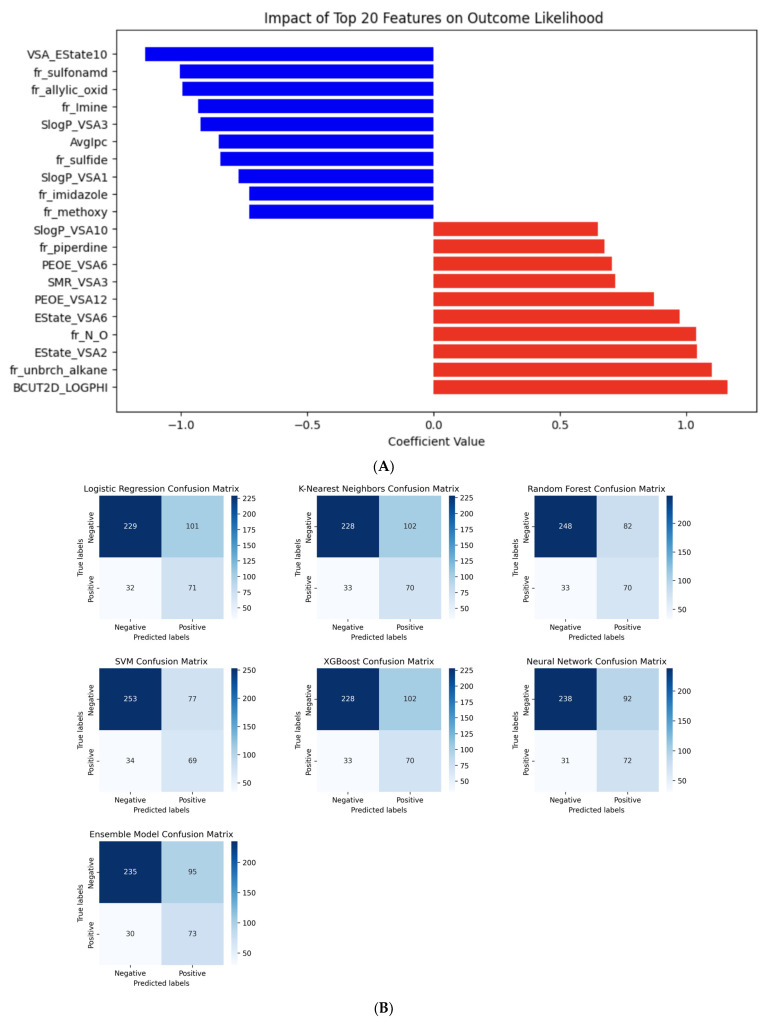
Confusion matrix for all models and important logR features. (**A**) Top 10 positive and negative logR features and their coefficient values. (**B**) Summary of model performance based on the confusion matrix (**C**) Top 20 RF important features and their correlation with active compounds. Color bars in Figure 2A and Figure 2C indicate features that have a strong positive (red) or negative correlation (blue) with the target variable (ACTIVITY). The correlation coefficient values reinforce that the positive (+1) or negative (−1) correlation is perfect.

**Figure 3 ijms-25-02646-f003:**
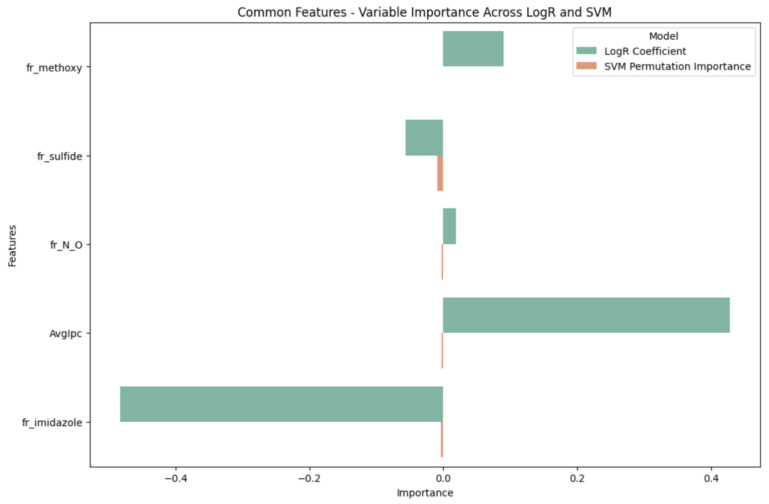
Common SVM and logR variable important features.

**Table 1 ijms-25-02646-t001:** Performance comparison of all models.

**S/N**	**SVM**				**XGB**				**NN**							
	**Training**	**Validation**	**Test**	**New (EVD)**	**Training**	**Validation**	**Test**	**New (EVD)**	**Training**	**Validation**	**Test**	**New (EVD)**				
**Recall**	0.87	0.87	0.88	0.67	1	0.86	0.91	0.68	0.73	0.92	0.91	0.7				
**Accuracy**	0.93	0.9	0.9	0.74	1	0.84	0.88	0.69	0.76	0.93	0.94	0.72				
**AUC–ROC**	0.93	0.91	0.9	**0.72**	1	0.84	0.88	0.69	0.76	0.97	0.96	**0.79**				
**F1-Score**	0.92	0.91	0.9	0.55	1	0.85	0.52	0.5	0.75	0.94	0.94	0.54				
**Cohen’s Kappa**	0.86	0.81	0.79	0.38	1	0.69	0.76	0.3	0.52	0.87	0.88	0.35				
	**Ensemble**				**LogR**				**KNN**				**RF**			
	**Training**	**Validation**	**Test**	**New (EVD)**	**Training**	**Validation**	**Test**	**New (EVD)**	**Training**	**Validation**	**Test**	**New (EVD)**	**Training**	**Validation**	**Test**	**New (EVD)**
	0.97	0.93	0.94	0.71	0.94	0.82	0.82	0.69	0.84	0.89	0.88	0.68	1	0.86	0.88	0.68
	0.97	0.87	0.87	0.71	0.95	0.79	0.78	0.69	0.86	0.83	0.82	0.69	1	0.84	0.84	0.73
	0.97	0.87	0.86	**0.71**	0.95	0.78	0.78	0.69	0.86	0.83	0.82	0.69	1	0.84	0.84	**0.72**
	0.97	0.89	0.88	0.54	0.95	0.8	0.79	0.51	0.86	0.85	0.83	0.51	1	0.85	0.85	0.55
	0.94	0.75	0.74	0.35	0.72	0.57	0.56	0.31	0.72	0.66	0.65	0.3	1	0.67	0.68	0.37

EVD—external validation data.

**Table 2 ijms-25-02646-t002:** Comparison of important features in SVM, logR, RF and XGB models.

S/N	SVM	logR	RF	XGB
1	fr_phenol	BCUT2D_LOGPHI	MaxAbsEstateIndex	SMR_VSA3
2	fr_phenol_noOrthoHbond	VSA_EState10	BCUT2D_MRHI	Chi2v
3	frAr_OH	fr_unbrch_alkane	MaxEStateIndex	frAr_OH
4	fr_N_O	Estate_VSA2	FpDensityMorgan2	fr_methoxy
5	fr_priamide	fr_N_O	FpDensityMorgan3	FpDensityMorgan3
6	PEOE_VSA11	fr_sulfonamd	PEOE_VSA6	HeavyAtomMolWt
7	SlogP_VSA7	fr_allylic_oxid	VSA_EState3	MaxAbsEstateIndex
8	fr_imidazole	Estate_VSA6	Chi2v	NumRotatableBonds
9	fr_piperdine	fr_Imine	MolWt	Chi4n
10	fr_C_S	SlogP_VSA3	PEOE_VSA9	PEOE_VSA13
11	BCUT2D_MWLOW	PEOE_VSA12	Chi1v	SMR_VSA7
12	fr_methoxy	Avglpc	MolWt	Fr_para_hydroxylation
13	BCUT2D_CHGLO	fr_sulfide	PEOE_VSA9	BCUT2D_MWLOW
14	PEOE_VSA3	SlogP_VSA1	Chi1v	MolMR
15	fr_sulfide	fr_imidazole	BCUT2D_MWHI	HallKierAlpha
16	lpc	fr_methoxy	ExactMolWt	MaxPartialCharge
17	Estate_VSA8	SMR_VSA3	Estate_VSA2	VSA_Estate9
18	Avglpc	fr_benzene	Kappa1	Estate_VSA2
19	fr_oxime	PEOE_VSA6	SlogP_VSA1	SMR_VSA1
20	fr_hdrzone	fr_ether	Estate_VSA8	fr_NHO

Color indications: Orange = structural descriptors, Green = electronic descriptors, Brown = physicochemical descriptors, and Purple = molecular descriptors.

## Data Availability

Data are contained within the article and Appendix A.

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
