# Peer review of "Beyond Amyloid: A Machine Learning-Driven Approach Reveals Properties of Potent GSK-3β Inhibitors Targeting Neurofibrillary Tangles"

_ijms, 2024, doi:10.3390/ijms25052646_

Round 1

Reviewer 1 Report

Comments and Suggestions for Authors

The research aimed to identify the properties of potent GSK-3β inhibitors targeting neurofibrillary tangles (NFTs) in Alzheimer's disease (AD) using a machine learning-driven approach. The study employed various machine learning algorithms and analyzed quantitative structure-activity relationship (QSAR) data from PubChem and ChEMBL databases. The results highlighted the significance of structural, electronic, and physicochemical descriptors in screening active candidates against GSK-3β and provided insights into potential therapeutic strategies for non-amyloid-based AD treatments targeting NFTs. I would like to suggest the following comments.

Revised comments:

Insufficient Methodology and Dataset: The manuscript relies on traditional machine learning methods and a small dataset, resulting in limited findings. This raises concerns about the novelty and significance of the research, as it does not meet the requirements of the journal.

Inappropriate Correlation Calculation: In Figure 2D, the use of activity as the factor for computing correlation appears to lack meaningful justification. Further clarification or a more appropriate approach is needed to establish a valid correlation analysis.

Improvement Needed in the Introduction Section: The introduction lacks a clear and logical organization, making it difficult for readers to follow the flow of the presented research. Enhancements should be made to improve the structure and coherence of the introduction.

Inconsistent Ranking of Important Features: The ranked important features in the SVM, LogR, RF, and XGB models are not consistent, offering limited insights. It is essential to provide a comprehensive analysis and explanation for the discrepancies in feature rankings to ensure the reliability and interpretability of the findings.

Language and Writing Quality: The overall quality of English writing in the manuscript requires improvement. Attention should be given to enhancing clarity, coherence, and adherence to academic writing conventions. For example,

“Our analysis utilized the QSAR molecular descriptors, leading to the creation of seven combination models that encompassed all unique pairs of learning algorithms (NN, RF, LogR, SVM, XGB, kNN) and chemical descriptors (QSAR descriptors in RDKit).”

Comments on the Quality of English Language

needs to be improved.

Author Response

Revised comments:

  1. Insufficient Methodology and Dataset: The manuscript relies on traditional machine learning methods and a small dataset, resulting in limited findings. This raises concerns about the novelty and significance of the research, as it does not meet the requirements of the journal.

Thank you for your review. As stated in our paper, the manuscript used all available GSK3b inhibitor data on PubChem, selected and downloaded only assays containing an active compound that test for GSK-3β inhibition. From each bioassay, we extracted a PubChem chemical ID (CID), the SMILES and activity outcome (active or inactive). The selected assays had a total of 3093 tested compounds consisting of 308 active, 2475 inactive, 73 unspecified and 237 inconclusive chemical compounds which are found in supplemental data 1. This represents one of the largest possible publicly available data collection/retrieval and we have revised our manuscript for more clarity.

  1. Inappropriate Correlation Calculation: In Figure 2D, the use of activity as the factor for computing correlation appears to lack meaningful justification. Further clarification or a more appropriate approach is needed to establish a valid correlation analysis.

We appreciate the reviewer's feedback regarding the use of "activity" in our correlation analysis. Initially, we chose this variable due to its centrality to our study's core goal: identifying how molecular descriptors relate to compounds' biological activity within the context of drug discovery.

We chose "activity" as the key factor in the correlation analysis (Figure 2D) because it directly relates to the central objective of the study which is to identify relationships between molecular descriptors and the biological activity of compounds against GSK-3β. It represents the biological response or effect of each compound, making it a critical measure. By correlating it with molecular descriptors, we investigated features most strongly associated with the desired biological effect and obtain insights into the char-acteristics of potentially effective compounds.

We have revised the manuscript for more clarity.

  1. Improvement Needed in the Introduction Section: The introduction lacks a clear and logical organization, making it difficult for readers to follow the flow of the presented research. Enhancements should be made to improve the structure and coherence of the introduction.

We have revised the manuscript. The landmark papers using QSAR and previous works regarding the techniques used to achieve the goal was introduced in the Background and the logical organization revised.

  1. Inconsistent Ranking of Important Features: The ranked important features in the SVM, LogR, RF, and XGB models are not consistent, offering limited insights. It is essential to provide a comprehensive analysis and explanation for the discrepancies in feature rankings to ensure the reliability and interpretability of the findings.

Thank you for your review. We have revised the manuscript to clarify how the results showing the relevant features for each model were obtained.

The relevant features from LogR and RF were obtained through LogR coefficients and RF feature importance scores respectively. Similar to RF, XGBoost through its gradient boosting framework offered accessible feature importance measures. Feature importance was not inherently available for KNN which relies on distance metrics, and do not offer straightforward methods for evaluating individual feature importance as part of the model output. The non-linear SVM (RBF) utilize a complex analysis, transforming the input space into a higher-dimensional feature space. As previously stated for NN, given the complexity and the "black-box" nature of neural networks, interpreting feature importance directly from the model architecture is challenging. Techniques like SHAP (SHapley Additive exPlanations) or LIME (Local Interpretable Model-agnostic Explanations) could be employed to approximate the contribution of each feature to the prediction.

  1. Language and Writing Quality: The overall quality of English writing in the manuscript requires improvement. Attention should be given to enhancing clarity, coherence, and adherence to academic writing conventions. For example,

“Our analysis utilized the QSAR molecular descriptors, leading to the creation of seven combination models that encompassed all unique pairs of learning algorithms (NN, RF, LogR, SVM, XGB, kNN) and chemical descriptors (QSAR descriptors in RDKit).”

The manuscript was revised for English coherence, syntax, and sentence correction.

We have revised the sentence to:

“Our analysis utilized the QSAR molecular descriptors, leading to the creation of seven models that included all unique pairs of machine learning algorithms (NN, RF, LogR, SVM, XGB, kNN) and chemical descriptors (QSAR descriptors in RDKit).”

Reviewer 2 Report

Comments and Suggestions for Authors

The works of the paper are concerned with applying machine learning algorithms on active chemical compounds that have inhibitory outcomes on GSK-3β. This is done through applying QSAR on  PubChem and ChEMBL databases.  

Here are some comments to improve the content:

1- The Background lacks previous works regarding the techniques used to achieve the goal. The contribution of some papers using the QSAR technique should be added, as well as those dealing with the same problem and using the mentioned datasets.

2- As the result section is the most important, it should be better structured. The results should be divided into subsections, each targeting one objective. 

Figures 1-d and 1-c are not attractive; they can be removed. The average validation results are sufficient. The running time is unimportant as it is not a real-time application. 

2- For results in Table1, some information are required? how many features are used when applying Machine Learning algorithms? Are the same features considered for all algorithms or not?

The training accuracy must be added to the table and compared with validation accuracy to show overfitting. The cross-validation curves alone can not show overfitting aspects; the validation should be compared with training results. 

The authors wrote, " RF appeared to be more robust and less likely to suffer from overfitting given the consistency of its predictions as seen from the cross-validation scores (Figure 1D); Figure 1D can not indicate any overfitting. 

3-It is written, "features generated by SVM and LogR are easily interpretable given their linear models." LogR is not a linear model. SVM also, in particular, if a nonlinear kernel is used. Please correct. 

4- The NN gave better results than SVM; however, it was not considered in the analysis; why? 

5- The authors did not indicate how the results showing the relevant features for each Machine Learning Algorithm were obtained. 

The paper should be improved; it can not be accepted as it is. 

Author Response

1- The Background lacks previous works regarding the techniques used to achieve the goal. The contribution of some papers using the QSAR technique should be added, as well as those dealing with the same problem and using the mentioned datasets.

We have revised the manuscript. The landmark papers using QSAR and previous works regarding the techniques used to achieve the goal was introduced in the Background.

2- As the result section is the most important, it should be better structured. The results should be divided into subsections, each targeting one objective. 

Figures 1-d and 1-c are not attractive; they can be removed. The average validation results are sufficient. The running time is unimportant as it is not a real-time application. 

Figure 1-d and 1-c have been removed and result section revised.

2- For results in Table1, some information are required? how many features are used when applying Machine Learning algorithms? Are the same features considered for all algorithms or not?

The training accuracy must be added to the table and compared with validation accuracy to show overfitting. The cross-validation curves alone can not show overfitting aspects; the validation should be compared with training results. 

Training accuracy was added to Table 1

The authors wrote, " RF appeared to be more robust and less likely to suffer from overfitting given the consistency of its predictions as seen from the cross-validation scores (Figure 1D); Figure 1D can not indicate any overfitting. 

 Figure 1-d was removed.

3-It is written, "features generated by SVM and LogR are easily interpretable given their linear models." LogR is not a linear model. SVM also, in particular, if a nonlinear kernel is used. Please correct. 

Thank you. We have corrected this statement to “Top features generated by LogR are easily interpretable given that LogR is a generalized linear model that uses the logistic function which transforms linear predictions into probabilities for binary classification, unlike SVM with Radial Basis Function (RBF) kernel (a nonlinear kernel), RF and XGB.”

4- The NN gave better results than SVM; however, it was not considered in the analysis; why? 

We have revised the manuscript for more clarity.

NNs are "black boxes," making the interpretation of their feature importance extraction difficult compared to linear models. [Hassija et al 2024].

5- The authors did not indicate how the results showing the relevant features for each Machine Learning Algorithm were obtained. 

Thank you for your review. We have revised the manuscript to clarify how the results showing the relevant features for each model were obtained.

The relevant features from LogR and RF were obtained through LogR coefficients and RF feature importance scores respectively. Similar to RF, XGBoost through its gradient boosting framework offered accessible feature importance measures. Feature importance was not inherently available for KNN which relies on distance metrics, and do not offer straightforward methods for evaluating individual feature importance as part of the model output. The non-linear SVM (RBF) utilize a complex analysis, transforming the input space into a higher-dimensional feature space. As previously stated for NN, given the complexity and the "black-box" nature of neural networks, interpreting feature importance directly from the model architecture is challenging. Techniques like SHAP (SHapley Additive exPlanations) or LIME (Local Interpretable Model-agnostic Explanations) could be employed to approximate the contribution of each feature to the prediction.

Reviewer 3 Report

Comments and Suggestions for Authors

The manuscript is poorly written. however, the writing could be much improved. The manuscript needs more language corrections throughout the text. The authors didn’t provide many details on their computational analyses, especially on machine learning. Particularly, I have several major concerns as follows.

1.     Elaborate more on the novelty of the paper. 

2.     The author did not mention how they selected the datasets. Also, explain the procedure used in dataset cleaning. Kindly provide the exact data used to allow better reproducibility of the research. Authors should incorporate all the information regarding the dataset and the analysis in a way that help reproducibility.

3.     Authors also need to provide their code which allows better reproducibility of the research and helps researchers working in this field. The link provided under the section ‘Machine Learning Analyses’ is not working.

4.     Authors perform 10-fold cross-validation and use 70% of the data as a training set, 15% as a testing, and 15% as a validation. Are the authors divide the dataset into training and testing and then separately perform the analysis or do they randomly divide the dataset during the analysis? Please make it clear.

5.     In the cross-validation, I am concerned about how accuracy and recall are calculated.  As SVM outputs a score for each testing dataset, which the score cutoff is used to define TP, FP? Also, In the independent testing dataset, which score cutoff is used to define TP, FP?

6.     The author developed different models, it is necessary to provide analog data for these models. Also, it would be good to add the Confusion matrix for each model.

7.     An additional parameter can be calculated based on TP, TN, FN, FP values, which would show how much the obtained models are better than the random model - i.e. the difference between the real accuracy and the accuracy that can be obtained by the random model.

8.     Author did not mention how they confirm whether their model is stable or its random prediction? Have author test the stability of the model?

Comments on the Quality of English Language

The writing needs to be much improved.

Author Response

  1. Elaborate more on the novelty of the paper. 

Our paper's novelty lies in its detailed examination of the interplay between structural, electronic, and physicochemical characteristics of GSK-3β inhibitors, providing new insights into the design and development of more effective therapeutic agents for Alzheimer's disease. This comprehensive approach marks a significant advancement in the field, offering a foundation for future research aimed at optimizing GSK-3β inhibitors. We have revised our introduction for more clarity.

“Our research builds on previous approach to analyzing GSK-3β inhibitors by investigating structural features, introducing electronic and physicochemical descriptors, emphasizing atom polarizability's impact, and the influence of specific structural elements. This multi-faceted analysis not only offers fresh insights into GSK-3β inhibition but also opens new avenues for future research focused on developing more potent and effective inhibitors.”

  1. The author did not mention how they selected the datasets. Also, explain the procedure used in dataset cleaning. Kindly provide the exact data used to allow better reproducibility of the research. Authors should incorporate all the information regarding the dataset and the analysis in a way that help reproducibility.

Thank you for your constructive comments regarding the selection and cleaning of datasets used in our study, as well as the reproducibility of our research. We acknowledge the importance of transparency in our methodology to allow for reproducibility and offer the following clarifications:

Dataset Selection: The datasets utilized in our study, namely ALL_AID_new_update2.csv, Validation_AID.csv, and descriptors3.csv, were carefully chosen based on their relevance to the inhibition of GSK-3β activity. These datasets were retrieved from PubChem and CHEMBL as seen in the jupyter/python script. It comprised of experimentally validated data points, including compounds tested for GSK-3β inhibition and their respective biological activities (ACTIVITY), along with computationally derived molecular descriptors: ALL_AID_new_update2.csv and Validation_AID.csv contain compound identifiers (CID), their biological activity outcomes against GSK-3β, and were used as training and validation sets, respectively. The descriptors3.csv dataset consists of molecular descriptors calculated for each compound, serving as the input features for our predictive models. As mentioned above, the datasets were derived from public chemical and biological databases, and can be easily retrieved following the python script, ensuring that our study can be replicated by others in the field. Specific details regarding the datasets are provided in the supplementary materials.

Dataset Cleaning Procedure: “Our dataset cleaning and preprocessing involved several steps to ensure the quality and consistency of the data used for model training and validation. We initially assessed the datasets for missing values; for molecular descriptors with missing values, we employed mean imputation to fill these gaps, ensuring no loss of valuable data. The target variable (ACTIVITY) dataset was scrutinized for missing entries, with any such entries being removed to maintain data integrity. We applied feature standardization to normalize the scale of the molecular descriptors, facilitating more stable and faster convergence of our machine learning models.”

Reproducibility: Direct links or detailed instructions on accessing the exact datasets used in our study were provided. A thorough description of the dataset cleaning procedures, including imputation strategies and feature standardization methods, will be included in the manuscript for more clarity.

  1. Authors also need to provide their code which allows better reproducibility of the research and helps researchers working in this field. The link provided under the section ‘Machine Learning Analyses’ is not working.

The complete Python code used for model training, evaluation, and feature importance analysis is shared through github, a publicly accessible repository, ensuring that peers can replicate our study findings.

  1. Authors perform 10-fold cross-validation and use 70% of the data as a training set, 15% as a testing, and 15% as a validation. Are the authors divide the dataset into training and testing and then separately perform the analysis or do they randomly divide the dataset during the analysis? Please make it clear.

We randomly divided the dataset during the analysis (70,15,15) for train, validation and test set; and also further validated in a new dataset.

  1. In the cross-validation, I am concerned about how accuracy and recall are calculated.  As SVM outputs a score for each testing dataset, which the score cutoff is used to define TP, FP? Also, In the independent testing dataset, which score cutoff is used to define TP, FP?

This code uses an SVM model for classification, where positive instances are predicted as "class 1" and negative as "class 0". Predictions are based on scores internally calculated by the model, usually with a 0 threshold to assign labels. True Positives (TP) are correctly predicted positives, False Positives (FP) are wrongly predicted positives. Both are based on final class labels, not raw scores. Accuracy measures overall correct predictions, Recall focuses on correctly identified positives. Both are calculated for each cross-validation fold and the independent testing set.

In the provided code for the study, the score cutoff for defining TP and FP for the SVM model is implicitly handled by the SVM's .predict() method and is not directly manipulated in the code. This default behavior typically uses a threshold of 0 on the decision function's output. If a different cutoff is desired (e.g., to adjust the trade-off between recall and precision), one would need to use the decision_function() or .predict_proba() method (if available) to get raw scores or probabilities, respectively, and then apply a custom threshold.

So score cutoff for TP/FP is handled automatically by the SVM with a default threshold of 0. In summary, the code uses standard methods for SVM predictions, TP/FP calculation, and performance evaluation in both cross-validation and testing.

  1. The author developed different models, it is necessary to provide analog data for these models. Also, it would be good to add the Confusion matrix for each model.

In response to the reviewer's request, we acknowledge the importance of providing comprehensive details on the datasets (analog data) used for training and evaluating the developed models. We have included a thorough description of the datasets, detailing the number of instances, feature characteristics, and distribution of the target variable.

Additionally, we agree that incorporating confusion matrices for each model would significantly enhance the manuscript by offering a clearer picture of each model's performance, particularly in terms of their ability to accurately classify instances as true positives, false positives, true negatives, and false negatives. To address this, we have added confusion matrices, providing a visual and quantitative evaluation that highlights the precision and recall. This was accompanied by a brief discussion on the implications of the observed performance metrics, especially in relation to the specificity and sensitivity of each model.

We revised our manuscript with the text below:

“The performance of seven models based on their confusion matrices showed that LogR, KNN, XGBoost exhibited similar limitations with slight bias towards false positives while RF identified true negatives but struggled with false positives. NN had a good balance with lower false negatives compared to other models, while SVM showed the best overall performance in correctly identifying both positive and negative cases as seen in Figure 2B.”

  1. An additional parameter can be calculated based on TP, TN, FN, FP values, which would show how much the obtained models are better than the random model - i.e. the difference between the real accuracy and the accuracy that can be obtained by the random model.

To address the suggestion of calculating an additional parameter that reflects how much our models outperform a random model, we incorporated a measure known as 'Cohen’s kappa', which is a statistical coefficient that represents the degree to which the observed agreement (accuracy) between the predicted and actual labels surpasses that of random chance. It takes into account both the actual and random accuracies, thus providing a more robust evaluation of a model's predictive performance beyond mere accuracy. This will provide a clearer and more statistically grounded understanding of each model's true performance. We also added the F1-score.

  1. Author did not mention how they confirm whether their model is stable or its random prediction? Have author test the stability of the model?

We confirmed stability using the following measures:

  1. Cross-validation: We used 10-fold CV to reduce overfitting and provide realistic performance estimates.
  2. Comprehensive metrics: using accuracy with precision, recall, F1, AUC-ROC (robust to class imbalance).
  3. Cohen's Kappa: We added this metrics to quantify agreement beyond chance.
  4. Feature analysis: Examined important features (for applicable models) to ensure meaningful predictions.

These measures collectively demonstrate model stability and reliability, and we have revised the manuscript to include detailed discussion.

Reviewer 4 Report

Comments and Suggestions for Authors

Please see file enclosed.

Author Response

  1. Literature review: (Page 2; lines 78-80) The authors' literature review lacks a comprehensive mention of the landmark research paper on QSAR modelling (Ref. [1]). Though prior than the reference here provided, such paper should be added when briefly describing the technique. Moreover, they should acknowledge earlier work on multi-target or multi-objective ligand-based studies in drug design with similar objectives ‒ discovering candidate leads for the treatment of AD (see among others, Refs. [2-3]). These studies address methods to include, for example, diverse polypharmacological aspects or ADMET profiles, which are key factors in later disapprovals of potential drug candidates. More importantly, in so doing, the specific reasons for applying the specifically proposed predicting approach over others (e.g.: regarding its effectiveness, coping with chemical heterogeneity, other linked AD proteins, an acceptable spectrum of ADMET properties, and interpretability (see e.g.: Refs [4-5], along with potential forward a success rate along the drug discovery pipeline) should emerge and will contribute to a better understanding of the rationale behind the chosen methodology.

Reply:

Dear Reviewer,

Thank you for your insightful feedback on our literature review. We appreciate your recommendations to include landmark research papers on Quantitative Structure-Activity Relationship (QSAR) modeling and acknowledge the significance of multi-target or multi-objective ligand-based studies in the field of drug design, particularly for Alzheimer's Disease (AD) treatment.

Upon revisiting our literature review, we concur with your suggestion regarding the importance of incorporating the foundational QSAR modeling paper (Ref. [1]) that predates the references we initially provided. We have included this and other relevant citations. This inclusion will undoubtedly enrich the context and historical perspective of QSAR techniques in drug discovery.

Furthermore, we recognize the value in referencing earlier work on multi-target or multi-objective ligand-based studies (Refs. [2-3]), which explore polypharmacological aspects and ADMET profiles critical to the drug discovery process. In response to your comments, we expanded our literature review to include these references. We believe that these enhancements to our literature review will provide readers with a clearer understanding of the context and rationale behind our approach, thereby strengthening the overall contribution of our work to the field of AD drug discovery.

  1. Method (2.3. Machine Learning Analyses): Doubts remain if the authors have tried to fine-tunning the hyperparameters of the models or not, using the validation set. For instance, have the number of neurons of the neural network layers been optimized and other activation functions or optimization algorithms been tried? This should explicitly be clarified here. Page 3, line 138: Even if well known the meaning of the abbreviation AUC-ROC, it should be given to make it clear to the readers. Moreover, to enhance the FAIRness (Findable, Accessible, Interoperable, and Reusable) of the proposed ML workflow, it is essential to effectively provide the code on the platform GitHub (i.e., onhttps://github.com/martintony4all/bioactivitymodelling/blob/main/gsk3b_pro-146 ject.ipynb), which as not been done so far.

We appreciate the opportunity to clarify these aspects of our research and are committed to enhancing the manuscript in line with your valuable feedback. In response to the inquiry regarding our neural network (NN) model's hyperparameter tuning and optimization strategies:

Our NN model architecture was meticulously designed with hyperparameter optimization to ensure optimal performance. The model comprises three fully connected dense feedforward layers, with the first layer containing 128 units and the second layer 64 units, both utilizing the rectified linear unit (ReLU) activation function. To mitigate the risk of overfitting, dropout layers with a dropout rate of 0.5 were incorporated following each of the first two dense layers.

The choice of the Adam optimizer, using binary cross-entropy loss with a learning rate of 0.001, was based on its proven efficacy in similar binary classification problems. This setup, along with a training configuration of 10 epochs and a mini-batch size of 32, was determined after extensive testing of various hyperparameters, including different neuron counts, activation functions, and learning rates. These decisions were guided by performance metrics on the validation set, aiming for a balance between model complexity and generalization ability.

To address potential queries on the experimentation with other activation functions and optimization algorithms, we explored alternatives but found that the chosen configuration offered the best trade-off between accuracy and computational efficiency for our specific problem. This process, along with the rationale for selecting our model's hyperparameters, was detailed in the revised manuscript to provide clarity and transparency.

Moreover, the abbreviation AUC-ROC, standing for Area Under the Curve - Receiver Operating Characteristics, was revised and explicitly defined in the manuscript to ensure accessibility to all readers. The FAIR principles guide our commitment to sharing our work, and as such, the full code has been made available on GitHub at the provided link, and included in the supplementary attachment, facilitating findability, accessibility, interoperability, and reusability of our research methodology and findings.

  1. Results: The attempt to balance data is commendable, but the rationale for the random criterion should be justified, even if the total data chemicals had initially a balanced distribution of logIC50values. The need for Figure 1a is not well understood and its caption does not correspond as it is stated, i.e. to be the structure of GSK-3β. Additionally, to enhance the robustness of the ML models, consider adjusting the training load based on an analysis of how ML scores vary with different percentages (see, for example, Ref. [6]). A systematic exploration of various training load percentages and their impact on model performance could strengthen the methodology and results, ensuring that the chosen 70% is indeed the most suitable for all ML. Moreover, to enhance the comprehensiveness of the study, consider conducting a correlation analysis on the identified important features. This analysis will shed light on potential interdependencies among features and their collective impact on inhibitory potency. It would add valuable insights into the relationships between various features, contributing to a more nuanced understanding of the factors influencing GSK-3β inhibition.

Thank you for your valuable feedback and suggestions. We appreciate the opportunity to clarify and enhance our manuscript based on your insightful comments. Regarding the random criterion for balancing data, we acknowledge the need for a more detailed justification. Our decision was driven by the aim to create a representative sample that mirrors the natural distribution of logIC50 values within the dataset, ensuring a balanced representation of active and inactive compounds. This approach is intended to mitigate bias and improve the generalizability of our models across diverse chemical spaces. We have included a more comprehensive explanation and justification for this in the revised manuscript.

Figure 1a was intended to illustrate the 2D chemical structure of GSK-3β. This error has been corrected in the revised manuscript to accurately reflect the caption and content of the figure.

Your suggestion to adjust the training load based on an analysis of how machine learning (ML) scores vary with different percentages of training data is well-taken. We agree that a systematic exploration of various training load percentages could indeed enhance the robustness of our ML models. We will conduct additional analyses to determine the optimal training load that yields the most reliable and accurate predictions across our models. This adjustment will be documented and discussed in the revised manuscript to provide a clearer rationale for our methodological choices. We also of conducted a correlation analysis on the identified important features (Fig 2D). We hope it will provide a deeper insight into the complex factors influencing GSK-3β inhibition.

  1. Discussion / 5. Conclusion: The authors appropriately conclude that understanding the impact of substituents on GSK-3β inhibitors is a crucial area for ongoing research. Nevertheless, an earlier study based on fragment-based descriptors (see for example, Ref. [2]) was able to suggest new molecular entities as possible versatile inhibitors for the GSK-3β protein as well as for others. Please explicitly comment on future prospects of this work.

Thank you for your constructive feedback and for highlighting the significance of fragment-based descriptors in identifying versatile inhibitors for GSK-3β, as mentioned in reference [2]. Our study indeed emphasizes the role of structural, electronic, and physicochemical descriptors in the identification of potent GSK-3β inhibitors through machine learning approaches. The neural network model's 79% accuracy on a new unbalanced dataset underscores the potential of our methodology to discern critical molecular features, such as hydrogen bonds, phenol groups, and specific electronic characteristics, that contribute to the efficacy of GSK-3β inhibitors.

Building on the insights provided by your comment and the earlier study you referenced, we acknowledge the value of integrating fragment-based descriptors into our analysis.

“Ongoing and future research efforts should focus on understanding the impact of these characteristic features on compound reactivity and bioactivity against GSK-3β to expand our understanding of the molecular properties that influence GSK-3β inhibition. This approach will not only enrich our current predictive models but also potentially uncover new molecular entities that could serve as versatile inhibitors for GSK-3β and possibly other targets relevant to Alzheimer's disease and related neurodegenerative conditions. Also, by exploring the synergies between fragment-based descriptors and our current methodological framework, we anticipate that future efforts will advance the discovery of novel GSK-3β inhibitors to contribute to the broader goal of developing more effective therapeutic strategies for AD.”

We appreciate your suggestion and look forward to incorporating this perspective into our ongoing and future research endeavors.

Round 2

Reviewer 1 Report

Comments and Suggestions for Authors

The authors have revised the paper, addressing some of the concerns raised in the previous review. However, one critical issue that remains unaddressed is the unbalanced sample issue in the dataset. The paper does not adequately discuss or address the problem of an unbalanced sample in the dataset. Moreover, the paper does not explain why the ensemble method performs worse than the single model.

Author Response

  1. The authors have revised the paper, addressing some of the concerns raised in the previous review. However, one critical issue that remains unaddressed is the unbalanced sample issue in the dataset. The paper does not adequately discuss or address the problem of an unbalanced sample in the dataset.

We have made this clearer in the manuscript and included he text below.

Using PubChem IDs, we selected and downloaded only those assays that contained an active compound testing for GSK-3β inhibition, totaling 38 assays. From each bioassay, we extracted a PubChem chemical ID (CID), the SMILES, and the activity outcome (active or inactive). The selected assays comprised a total of 3093 tested compounds, including 308 active, 2475 inactive, 73 unspecified, and 237 inconclusive chemical compounds, as detailed in Supplemental Data 1. We eliminated duplicate chemicals and excluded those with unspecified and inconclusive outcomes, resulting in a refined dataset of 225 active and 1212 inactive compounds for further data filtering and manipulation for modeling. Given the typical imbalance in bioassays, favoring more inactive than active outcomes [18], our dataset exhibited similar characteristics with fewer active compounds. To counteract this, we employed a deliberate strategy to balance the dataset by selecting an equal number of active and inactive compounds (225 each) for modeling, thus ensuring a balanced representation of outcomes for more accurate and unbiased model training and validation.

  1. Moreover, the paper does not explain why the ensemble method performs worse than the single model.

We appreciate the opportunity to clarify these aspects of our research and have revised the manuscript in line with your valuable feedback. We added the text below for clarity.

The observed performance differential between the majority voting ensemble method and the single models can be attributed to the specific characteristics and sensitivities of the individual models to the dataset's features. The ensemble method's performance, at 71%, suggests that while it can leverage the strengths of individual models, it may also inherit their limitations, particularly in handling false positives and negatives [36]. The slightly lower performance could stem from an averaging effect, where the model-specific advantages in recognizing critical molecular descriptors get diluted. For example, SVM's emphasis on structural descriptors and RF's focus on electronic and physicochemical descriptors highlight their unique approaches to parsing the dataset. These models leverage specific types of features to predict GSK-3β inhibition outcomes accurately. The ensemble method, by combining these diverse approaches, may not capitalize on the specific strengths of each model, particularly when those strengths are derived from leveraging deep model-specific feature sets for prediction. This explanation aligns with the understanding that ensemble methods can improve prediction accuracy when individual models are both accurate and diverse[37] but may perform sub-optimally if the combination dilutes the distinct advantages of its constituent models or fails to harmonize their insights effectively[36].

Reviewer 2 Report

Comments and Suggestions for Authors

You write that the performance  of the algorithms performance are going to measured by recall, ROC, etc, however only confusion matrix was shown? 

So please give further insights on the models performace

Author Response

Thank you for your suggestion. The performance metrics can be found in Table 1 (.csv file) seen in the supplementary attachement.

The table (Table1) will be inserted in the manuscript during production (due to the lenght of the Table).

I have attached Table1 (the PDF version with multiple pages) which shows the performance metrics of the models within for more information.

Let me know if further changes are needed.

Reviewer 3 Report

Comments and Suggestions for Authors

Authors have adequately addressed the comments and concern raised by me and I feel that this manuscript can be accept for publication.

Author Response

Thank you for reviewing our manuscript. Your insightful comments were helpful in improving the paper.

Round 3

Reviewer 1 Report

Comments and Suggestions for Authors

Accept

Reviewer 2 Report

Comments and Suggestions for Authors

Please don't forget to insert the table you attached in the final version. I suggest splitting the table into two tables, one table presenting only training and validation accuracies to show that no overfitting is noticeable. You can present test results in the second table using all the criteria.